# Intranasal Administration of Rotenone Reduces GABAergic Inhibition in the Mouse Insular Cortex Leading to Impairment of LTD and Conditioned Taste Aversion Memory

**DOI:** 10.3390/ijms22010259

**Published:** 2020-12-29

**Authors:** Hiroki Toyoda, Ayano Katagiri, Takafumi Kato, Hajime Sato

**Affiliations:** Department of Oral Physiology, Osaka University Graduate School of Dentistry, Suita 565-0871, Japan; ayano@dent.osaka-u.ac.jp (A.K.); takafumi@dent.osaka-u.ac.jp (T.K.)

**Keywords:** rotenone, insular cortex, taste aversion, GABA, LTD

## Abstract

The pesticide rotenone inhibits mitochondrial complex I and is thought to cause neurological disorders such as Parkinson’s disease and cognitive disorders. However, little is known about the effects of rotenone on conditioned taste aversion memory. In the present study, we investigated whether intranasal administration of rotenone affects conditioned taste aversion memory in mice. We also examined how the intranasal administration of rotenone modulates synaptic transmission and plasticity in layer V pyramidal neurons of the mouse insular cortex that is critical for conditioned taste aversion memory. We found that the intranasal administration of rotenone impaired conditioned taste aversion memory to bitter taste. Regarding its cellular mechanisms, long-term depression (LTD) but not long-term potentiation (LTP) was impaired in rotenone-treated mice. Furthermore, spontaneous inhibitory synaptic currents and tonic GABA currents were decreased in layer V pyramidal neurons of rotenone-treated mice compared to the control mice. The impaired LTD observed in pyramidal neurons of rotenone-treated mice was restored by a GABA_A_ receptor agonist muscimol. These results suggest that intranasal administration of rotenone decreases GABAergic synaptic transmission in layer V pyramidal neurons of the mouse insular cortex, the result of which leads to impairment of LTD and conditioned taste aversion memory.

## 1. Introduction

Rotenone is a highly specific inhibitor of mitochondrial complex I that originates from the roots of plants [1]. As rotenone is extremely lipophilic, it can easily cross the cell membrane without the assistance of a specific transport system [2]. Importantly, rotenone can easily pass through the blood–brain barrier, reaches the brain, and accumulates in mitochondria where it causes impairment of oxidative phosphorylation by inhibiting complex I and a decrease in energy production, which leads to neuronal death [3]. Evidence has shown that chronic treatment of animals with rotenone-induced selective degeneration of dopaminergic neurons and terminals in the substantia nigra pars compacta and striatum, respectively, motor deficits and postural imbalance reminiscent of Parkinson’s disease (PD) [4,5,6,7]. However, the neurotoxic effects of rotenone are not specific to the dopaminergic system. For example, it has been reported that the intravenous injection of rotenone in rats caused a loss of striatal serotoninergic and dopaminergic fibers, nigral dopaminergic neurons, striatal cholinergic interneurons, cholinergic neurons in the pedunculopontine tegmental nucleus, and noradrenergic neurons in the locus coeruleus [8]. The chronic administration of rotenone produces not only motor deficits including rigidity, postural instability, akinesia/bradykinesia and tremor, but also extranigral signs and non-motor symptoms such as gastrointestinal dysfunction, hyposmia, sleep disturbance, circadian dysfunction, cognitive impairment, anxiety, and depression [6,7,9]. However, there is no study to examine the effects of rotenone on conditioned taste aversion memory.

The neurotoxic effects of rotenone have been identified by using various routes of administration, and the most common route of administration is through a systemic administration including subcutaneous, intravenous, and intraperitoneal injections. In general, exposure of environmental chemicals such as rotenone occurs through inhalation, cutaneous contact and ingestion. Chemicals absorbed at the nasal mucosa reach the brain directly through an active axonal transport of olfactory neurons and/or passive diffusion to the cerebrospinal fluid [10,11]. Previous reports have shown that the intranasal administration of rotenone caused olfactory dysfunction and degeneration of dopaminergic fibers in the rat olfactory bulb and substantial nigra [12,13], but little is known about the effects of the intranasal administration of rotenone on neurotoxic effects.

The insular cortex plays essential roles in sensory perception including somatosensation and taste, emotion processing, motor control, and self-awareness [14,15,16], and has been accepted as pivotal brain area underlying taste learning, including conditioned taste aversion [17,18,19,20,21]. Long-term synaptic potentiation (LTP) and depression (LTD) are well recognized to be critical synaptic mechanisms associated with taste learning and memory [21,22]. Considering that it has been demonstrated that exposure of rotenone impaired LTP in the rat hippocampal slices [23], rotenone may cause impairment of conditioned taste aversion memory through modulating synaptic plasticity in the insular cortex.

In the present study, we first examined the effects of intranasal administration of rotenone on conditioned taste aversion memory. Furthermore, we examined how the intranasal administration of rotenone modulates synaptic transmission and plasticity in layer V pyramidal neurons of the mouse insular cortex. We found that the intranasal administration of rotenone impaired conditioned taste aversion memory to bitter taste in mice. We also found that the intranasal administration of rotenone impaired GABAergic synaptic transmission and LTD without affecting glutamatergic synaptic transmission and LTP. Our data indicate that the intranasal administration of rotenone decreases GABAergic synaptic transmission in layer V pyramidal neurons of the mouse insular cortex, which causes impairments of LTD and conditioned taste aversion memory.

## 2. Results

### 2.1. Conditioned Taste Aversion Memory is Impaired in Rotenone-Treated Mice

We first examined whether the intranasal administration of rotenone affects conditioned taste aversion memory to quinine hydrochloride (QHCl) in mice. Conditioned taste aversion is the most established form of taste learning and memory in animals [24,25]. When animals learn to associate a novel taste (conditioned stimulus, CS) with gastrointestinal malaise produced by LiCl (unconditioned stimulus, US), an association learning between the ingested substance and negative visceral consequences is quickly established. The experimental schedule performed in this study is shown in Figure 1A. The conditioned taste aversion was induced following pairing of CS (0.03 mM QHCl solution intake) with US (0.6 M LiCl injection). The intakes of QHCl solution after LiCl injection were significantly affected by treatment (DMSO and rotenone) (F_1,35_ = 22.1, *p* < 0.001) and time (F_6,35_ = 5.5, *p* < 0.001) but no significant interaction was found (F_6,35_ = 0.8, *p* = 0.601). After 1 day of LiCl injection (on the 15th day), conditioned taste aversion was obtained following pairing of CS with US both in vehicle-treated mice [Two-way ANOVA LSD: *p* = 0.027 vs before LiCl injection (on the 12th day), *n* = 6] and rotenone-treated mice [Two-way ANOVA LSD: *p* < 0.001 vs before LiCl injection (on the 12th day), *n* = 6] (Figure 1B). After 2 days of LiCl injection (on the 16th day), conditioned taste aversion was rapidly extinguished in rotenone-treated mice [Two-way ANOVA LSD: *p* = 0.239 vs. before LiCl injection (on the 12th day), *n* = 6] while it was maintained in vehicle-treated mice [Two-way ANOVA LSD: *p* = 0.048 vs before LiCl injection (on the 12th day), *n* = 6] (Figure 1B). After 2, 3, 4, and 5 days of LiCl administration (on the 16th, 17th, 18th, and 19th day), the amount of intake was significantly greater in rotenone-treated mice than in vehicle-treated mice (Two-way ANOVA LSD: on the 16th day; *p* = 0.049, on the 17th day; *p* = 0.028, on the 18th day; *p* = 0.026, on the 19th day; *p* = 0.011) (Figure 1B). These results suggest the intranasal administration of rotenone impairs conditioned taste aversion memory in mice.

### 2.2. Intranasal Administration of Rotenone Impairs LTD but Not LTP in Layer V Pyramidal Neurons of the Mouse Insular Cortex

Emerging evidence indicates that LTP and LTD in the insular cortex are cellular mechanisms of conditioned taste aversion memory [21]. Thus, we next investigated whether and how the intranasal administration of rotenone modulates LTP and LTD in layer V pyramidal neurons of the insular cortex using vehicle-treated and rotenone-treated mice. Whole-cell patch-clamp recordings were performed from visually identified layer V pyramidal neurons. The typical firing pattern of pyramidal neurons displayed firing frequency adaptation [26]. After identification of pyramidal neurons, LTP-inducing stimuli were applied. For induction of LTP, 80 presynaptic stimuli were delivered at 2 Hz while recorded neurons were held at +30 mV for the duration (40 s) of the LTP-inducing presynaptic stimuli [26]. The LTP-inducing stimuli caused a significant enhancement of synaptic responses in both vehicle-treated mice (25–30 min after the LTP-inducing stimuli, 150.0 ± 12.2% of baseline responses, *n* = 8; paired *t*-test: t(7) = −4.31, *p* = 0.004) and rotenone-treated mice (25–30 min after the LTP-inducing stimuli, 148.5 ± 9.5% of baseline responses, *n* = 8; paired *t*-test: t(7) = −4.74, *p* = 0.002) (Figure 2A–C). There was no significant difference between the magnitudes of synaptic potentiation obtained from two groups (unpaired *t*-test: t(14) = 0.092, *p* = 0.928) (Figure 2C).

We next investigated whether the intranasal administration of rotenone affects LTD in layer V pyramidal neurons of the mouse insular cortex. The induction protocol for LTD involved 300 presynaptic pulses at 1 Hz while recorded neurons were held −45 mV for the duration (300 s) of the LTD-inducing presynaptic stimulation [27]. The LTD-inducing stimuli produced a significant depression of synaptic responses in layer V pyramidal neurons of vehicle-treated mice (25–30 min after the LTD-inducing stimuli, 73.1 ± 6.0% of baseline responses, *n* = 7; paired *t*-test: t(6) = 3.82, *p* = 0.009), while a depression of synaptic responses was not induced by the LTD-inducing stimuli in rotenone-treated mice (25–30 min after the LTD-inducing stimuli, 95.1 ± 4.5% of baseline responses, *n* = 7; paired *t*-test: t(6) = 1.45, *p* = 0.197) (Figure 2D–F). These results suggest that intranasal administration of rotenone impairs LTD but not LTP in layer V pyramidal neurons of the mouse insular cortex.

### 2.3. Intranasal Administration of Rotenone Affects Inhibitory Synaptic Transmission

We next examined whether the intranasal administration of rotenone affects excitatory and inhibitory synaptic transmission in layer V pyramidal neurons of the mouse insular cortex. The frequency and amplitude of sEPSCs recorded from layer V pyramidal neurons of rotenone-treated mice (frequency: 2.1 ± 0.3 Hz, amplitude: 15.7 ± 1.1 pA; *n* = 14) were comparable (frequency: unpaired *t*-test, t(26) = −0.22, *p* = 0.827; amplitude: unpaired *t*-test, t(26) = −0.18, *p* = 0.860) to those obtained from vehicle-treated mice (frequency: 2.2 ± 0.2 Hz, amplitude: 15.9 ± 0.8 pA; *n* = 14) (Figure 3A–C). However, the frequency of sIPSCs recorded from layer V pyramidal neurons of rotenone-treated mice (5.3 ± 0.4 Hz, *n* = 13) was significantly smaller (unpaired *t*-test: t(26) = 2.60, *p* = 0.015) than that of vehicle-treated mice (7.0 ± 0.5 Hz, *n* = 15), though the amplitude of sIPSCs recorded from layer V pyramidal neurons of rotenone-treated mice (20.9 ± 0.9 pA, *n* = 13) was not significantly different (unpaired *t*-test: t(26) = 0.39, *p* = 0.697) from that of vehicle-treated mice (21.5 ± 1.1 pA, *n* = 15) (Figure 3D–F). These results suggest that the intranasal administration of rotenone reduces inhibitory synaptic transmission without affecting excitatory synaptic transmission.

### 2.4. The Intranasal Administration of Rotenone Reduces Tonic GABA Currents and Activation of GABA_A_ Receptors Restores LTD in Rotenone-Treated Mice

It has been reported that GABA levels in the striatum of rotenone-treated rats were significantly reduced compared with those of the control rats [28]. Therefore, we next investigated whether tonic GABA currents are altered by the intranasal administration of rotenone. The inward shift of the baseline current caused by 10 μM bicuculline obtained from pyramidal cells of rotenone-treated mice (24.2 ± 2.1 pA, *n* = 7) was significantly smaller (unpaired *t*-test: t(13) = –4.51, *p* < 0.001), compared to that obtained from those of vehicle-treated mice (40.0 ± 2.7 pA, *n* = 8) (Figure 4A,B). This result suggests that the intranasal administration of rotenone reduces tonic GABA currents in layer V pyramidal neurons of the mouse insular cortex.

Considering that both phasic and tonic GABA currents are reduced in pyramidal neurons of rotenone-treated mice, it is highly possible that the impairment of LTD by rotenone is caused by reduced GABAergic activities. Then, we tested whether activation of GABA_A_ receptors by muscimol during LTD-inducing stimuli restores LTD in pyramidal neurons of rotenone-treated mice. When 50 μM muscimol was applied during LTD-inducing stimuli, LTD was restored (25–30 min after the LTD-inducing stimuli, 69.8 ± 5.4% of baseline responses, *n* = 8; paired *t*-test: t(7) = 6.00, *p* < 0.001) (Figure 4C). The magnitude of synaptic depression obtained in the presence of muscimol (Figure 4C) was significantly larger (unpaired *t*-test: t(13) = 3.57, *p* = 0.003), compared to that obtained without muscimol (Figure 2E,F). Although the impairment of LTD in rotenone-treated mice was restored by muscimol, there is a possibility that the restored LTD may have been caused by mainly the effect of the muscimol application rather than the LTD-inducing protocol (see [29]). Thus, we further examined the effect of muscimol application alone without LTD-inducing stimuli. We found that 50 μM muscimol application alone did not induce LTD in pyramidal neurons of rotenone-treated mice (25–30 min after muscimol application, 99.6 ± 5.7% of baseline responses, *n* = 7; paired *t*-test: t(6) = −0.37, *p* = 0.725) (Figure 4D), indicating that the effect of muscimol is transient and the restoration of LTD is required for not only muscimol but also LTD-inducing stimuli. Collectively, these results suggest the intranasal administration of rotenone reduces GABAergic transmission in the mouse insular cortex, leading to an impairment of LTD.

### 2.5. Intranasal Administration of Rotenone Has no Effect on Spike Firing Properties

It has been reported that rotenone suppressed spike firings of rat locus coeruleus neurons and mouse hippocampal neurons [30,31]. Thus, we investigated whether and how the intranasal administration of rotenone affects intrinsic membrane properties and spike firing properties under the current-clamp condition. There was no significant difference (unpaired *t*-test: t(28) = −0.21, *p* = 0.836) in resting membrane potential between vehicle-treated mice (−68.0 ± 0.9 mV, *n* = 16) and rotenone-treated mice (−67.7 ± 0.8 mV, *n* = 14). There was also no significant difference (unpaired *t*-test: t(28) = 0.55, *p* = 0.584) in input resistance between vehicle-treated mice (136.8 ± 9.5 MΩ, *n* = 16) and rotenone-treated mice (129.7 ± 8.0 MΩ, *n* = 14). To explore spike firing properties, voltage responses to depolarizing current pulses were recorded from layer V pyramidal neurons in vehicle-treated and rotenone-treated mice (Figure 5A,B). As shown in the relationship between current intensity and number of spikes (Figure 5C), there was no significant difference in number of spikes at all current intensities between vehicle-treated and rotenone-treated mice (unpaired *t*-test: 30 pA, t(28) = 0.93, *p* = 0.359; 60 pA, t(28) = −0.35, *p* = 0.729; 90 pA, t(28) = 0.11, *p* = 0.910; 120 pA, t(28) = 0.34, *p* = 0.735; 150 pA, t(28) = 0.10, *p* = 0.925; 180 pA, t(28) = −0.12, *p* = 0.902; 210 pA, t(28) = −0.25, *p* = 0.808; 240 pA, t(28) = −0.56, *p* = 0.579). These results suggest that the intranasal administration of rotenone has almost no effects on the intrinsic and spike firing properties.

## 3. Discussion

The present study, for the first time, revealed that the intranasal administration of rotenone impaired conditioned taste aversion memory to bitter taste in mice. We found that the intranasal administration of rotenone impaired inhibitory synaptic transmission without affecting excitatory synaptic transmission in layer V pyramidal neurons of the mouse insular cortex. We also found that the intranasal administration of rotenone impaired LTD but not LTP. These observations indicate that the intranasal administration of rotenone reduces GABAergic synaptic transmission in pyramidal neurons of the mouse insular cortex, which leads to an impairment of LTD and conditioned taste aversion memory. Our findings may be helpful for understanding the mechanisms of cognitive deficits caused by rotenone.

Rotenone is an interest subject investigation because it reproduces the features of human PD in animals. In humans, the most common risks of exposure to rotenone include inhalation, cutaneous contact, and ingestion. It is believed that rotenone has greater toxicity when inhaled rather than ingested because when inhaled, rotenone stays longer in the body due to its small particle, while, when ingested, gastrointestinal irritation after ingestion induces prompt vomiting responses [32]. However, neurotoxic effects of inhaled rotenone have not been well explored, whereas it has been established that systemic injection of rotenone in animals is able to induce not only motor deficits but also non-motor symptoms such as olfactory dysfunction, sleep disorders, and cognitive impairment [6]. Several studies have investigated the effects of rotenone on cognitive function. In rotenone-treated rats, transfer latency in elevated plus maze test was significantly prolonged compared with the controls, suggesting that cognitive decline is brought about by rotenone [33]. The cognitive impairment in rotenone-treated rats was also shown in a previous study using the novel object-recognition test [34]. In contrast, one study showed that the intragastric administration of rotenone for three months improved spatial learning and memory abilities in mice [35]. Thus, the effects of rotenone on cognitive functions are still a matter of debate. In the present study, we have shown that the intranasal administration of rotenone impaired conditioned taste aversion memory, supporting the deleterious role of rotenone on cognitive functions. However, more studies are necessary to establish the precise roles of rotenone on cognitive functions, including conditioned taste aversion memory.

At the synaptic level, it has been shown that acute application of rotenone reduced a field potential amplitude in the rat striatal spiny neurons, the result of which was associated with the development of membrane depolarization and inward currents [36]. A single systemic administration of rotenone did not cause neurotoxicity, but rather enhanced glutamate-mediated dopamine release in the rat striatum [37]. In rotenone-treated mice with subcutaneous injection, the amounts of glutamate and glutamine were significantly increased compared to sham-treated mice [38]. In contrast to these findings, our results showed that the intranasal administration of rotenone had almost no effects on glutamatergic synaptic inputs onto layer V pyramidal neurons of the mouse insular cortex. This is presumably because the intranasal administration of rotenone did not induce neurotoxicity in glutamatergic neurons of the insular cortex, similar to the observation made in glutamatergic neurons of the rat midbrain neuronal cultures following long-term exposure (12 h) of rotenone [39]. In addition, it has been demonstrated that the chronic intravenous treatment of rats with rotenone produced degeneration in the basal ganglia and brain stem nuclei, whereas it had little effect on the cerebellum, hippocampus and cerebral cortex [8]. Thus, it is likely that the pyramidal cell itself in the insular cortex is not easily degenerated by rotenone. Indeed, we have demonstrated that the intranasal administration of rotenone had almost no effects on the intrinsic and spike firing properties in pyramidal neurons of the insular cortex.

Unlike the unaltered glutamatergic synaptic transmission, GABAergic synaptic transmission was significantly decreased in the rotenone-treated mice compared with the vehicle-treated mice. It has been reported that the intranasal administration of rotenone induced dopaminergic neurite degeneration in the mouse olfactory bulb and reduced the inhibitory inputs onto mitral cells [13]. Because dopamine neurons in the glomerular layer of the olfactory bulb are inhibitory neurons for the mitral cells [40], these data may suggest that the reduced inhibitory inputs by rotenone are caused by the degeneration of dopamine neurons [13]. The dopaminergic afferents from the substantia nigra terminate on GABAergic interneurons in the rat insular cortex [41], and D1 and D2 dopamine receptors are substantially expressed in the cat and rat insular cortex [42,43]. Since it has been shown that the intranasal administration of rotenone caused degeneration of the substantia nigra dopaminergic neurons, the rotenone-induced reduction of inhibitory synaptic inputs onto insular layer V pyramidal neurons is possibly induced by the degeneration of the substantia nigra dopaminergic neurons. We have also found that tonic GABA currents were apparently decreased in rotenone-treated mice compared with the vehicle-treated mice. Since it has been found that administration of rotenone remarkably decreased GABA levels in the rat striatum [28], GABA levels might become lower in the insular cortex of the rotenone-treated mice.

Based on our results, the intranasal administration of rotenone produced almost no effect on glutamatergic synaptic transmission, while reducing GABAergic synaptic transmission in the mouse insular cortex. It is currently unclear whether the effect of the intranasal administration of rotenone is more specific to GABAergic neurons than glutamatergic neurons. The vulnerability to rotenone has previously been shown to be almost similar between cultured GABAergic and glutamatergic neurons [39,44]. However, GABAergic neurons appeared less elongated after 100 nM rotenone treatment, and thus it is indicated that GABAergic neurons are more susceptible to rotenone-induced oxidative stress [44]. In addition, rotenone has been shown to cause nitrosative stress [45], and it has been demonstrated that GABAergic neurons that express neuronal nitric oxidase synthase (nNOS) are more susceptible to toxic exposure due to their intrinsic high basal levels of nitrosative stress in comparison with glutamatergic neurons [46]. Thus, it is possible that rotenone may cause functional impairment in nNOS-expressing GABAergic neurons more easily than in glutamatergic neurons. Further studies of the precise mechanisms as to how the intranasal administration of rotenone affects GABAergic and glutamatergic neurons in the insular cortex would be necessary.

The insular cortex is a critical brain region that is associated with the acquisition, storage, and extinction of conditioned taste aversion memory [21]. LTP and LTD in the insular cortex are believed to be the cellular mechanisms of taste learning and memory [21,25]. In the present study, we found that the intranasal administration of rotenone impaired LTD without affecting LTP. Whereas it is widely accepted that long-term taste memory storage requires LTP-like mechanisms in the insular cortex [19,47], the role of LTD in taste learning and memory has been less investigated. It has been shown that induction of LTD in the basolateral amygdaloid nucleus to insular cortex pathway before conditioned taste aversion training enhanced its extinction [48]. In addition, acid-sensing ion channel (ASIC)-dependent LTD in the insular cortex has been shown to be required for extinction but not acquisition or retention of conditioned taste aversion memory [49]. These findings indicate that LTD in the insular cortex plays a role in extinction of conditioned taste aversion memory. On the other hand, our present findings suggest that LTD in the insular cortex is necessary for retention of conditioned taste aversion memory. How does LTD in the insular cortex contribute to retention of conditioned taste aversion memory? Memory formation is associated with strengthening of a subset of synapses [50,51], which may occur at the expense of weakening other synapses to maintain the overall synaptic strength balanced in the brain. Hence, LTD in the insular cortex may be required for modifying the local neural circuit in order to retain conditioned taste aversion memory.

Our data suggest that the decreased GABAergic synaptic transmission by rotenone is likely to cause impairment of LTD because activation of GABA_A_ receptors by muscimol during LTD-inducing stimuli restored LTD in pyramidal neurons of the rotenone-treated mice. Importantly, while our finding shows that the GABAergic synaptic transmission is reduced by rotenone, the impairment of LTD is brought about by an alteration of glutamatergic synaptic transmission. In other words, GABA inhibition is involved in modulation of LTD of glutamatergic synapses. The GABAergic inhibition has been shown to play a critical role in induction of glutamatergic LTD in several brain regions [29,52,53,54]. For instance, it has been demonstrated that glutamatergic LTD induced by low frequency stimulation was abolished by GABA_A_ receptor antagonists in the mouse amygdala [52] and the rat hippocampus [54]. These studies postulate that local feed-forward and/or feed-back circuitry mediated by GABAergic interneurons might be involved in induction of LTD. In the neocortex, fast-spiking basket interneurons primarily mediate feed-forward and feed-back inhibition within layer V [55]. Through this circuit, LTD of glutamatergic synapses might be induced in layer V pyramidal neurons of the mouse insular cortex. Thus, the reduction of dendritic GABAergic inhibition in rotenone-treated mice appears to shift the threshold for induction of LTD, resulting in an impairment of LTD. In addition, it has been shown that application of muscimol induced LTD in the rat hippocampus [53,54], the nucleus accumbens of adolescent mice [29], and the dorsolateral striatum of adult mice [29]. With regard to a possible mechanism for this type of synaptic modification, both presynaptic and postsynaptic mechanisms are likely to be involved. In particular, it has been shown that following application of muscimol, the release of endocannabinoids, which mediate retrograde signals occurred presynaptically to reduce release of glutamate in the striatum complex in an age- and region-dependent manner [29]. Thus, muscimol-mediated LTD is believed to be caused by retrograde messengers. However, we have demonstrated that muscimol application alone did not cause LTD in insular pyramidal neurons of rotenone-treated mice. Therefore, the latter possibility seems to be unlikely.

Previous pharmacological studies using GABA_A_ receptor agonist and antagonist demonstrated that GABAergic activity modulated acquisition and retrieval of conditioned taste aversion memory [21]. For instance, microinjection of muscimol impaired acquisition of conditioned taste aversion memory while that of bicuculline impaired its latent inhibition [56]. On the other hand, acquisition of conditioned taste aversion memory was not affected by the infusion of muscimol or bicuculline into the nucleus basalis magnocellularis [57]. Furthermore, it has been shown that activation of GABA_A_ receptors in the insular cortex disrupted taste memory formation but did not elicit a deficit of taste memory retrieval, whereas inactivation of GABA_A_ receptors did not impede novel memory formation but disrupted memory retrieval [58]. Thus, the consensus about the role of GABA in conditioned taste aversion memory is not established. Although our data suggest that GABA might be critical for retention of conditioned taste aversion memory, further studies would be necessary to clarify how GABA is involved in acquisition, consolidation, and retrieval of taste learning and memory.

In summary, our findings suggest that decreased GABAergic inhibition caused by the intranasal administration of rotenone is responsible for the impairment of LTD and conditioned taste aversion memory. These synaptic events may help to assist an understanding of the motor and non-motor deficits caused by rotenone.

## 4. Materials and Methods

All experiments were performed according to the European Communities Council Directive of 2010/63/EU. All animal experiments were approved by the Animal Ethics Committee of Osaka University Graduate School of Dentistry for the care and use of laboratory animals (Project identification code: 29-001-2; Date of approval: April 20th, 2017), and the experiments were conducted in accordance with the guidelines. Every effort was made to reduce the number and suffering of animals.

### 4.1. Animals

We used male C57BL/6J mice at 6-7 weeks of age (Japan SLC, Hamamatsu, Japan). The mice were caged individually in plastic cages (125 (W) × 199 (D) × 113 (H) mm; Japan Clea, Tokyo, Japan) and maintained at 23 ± 2 °C under 12-h light: 12-h dark cycles. Deionized water and food (Oriental Yeast, Tokyo, Japan) were available *ad libitum*. Rotenone (Sigma–Aldrich, St. Louis, MO, USA) was dissolved in 100% dimethyl sulfoxide (DMSO; Wako Pure Chemical Industries, Osaka, Japan) to make stock solution at concentrations of 0.1 M, and thereafter the stock solution was diluted by water. The mice were anesthetized with isoflurane (Sigma–Aldrich), and then the polyethylene micro tube (ID: 0.2 mm, OD: 0.4 mm, Length: 8 mm) that is connected to a peristaltic pump was put into the right side of the nose cavity. Rotenone solution (0.35 mg/kg) or vehicle solution (3.5% DMSO) was intranasally injected into the right side of the nose cavity for 7 days. The dose of rotenone was determined based on previous reports [12,13], to minimize the influence of a liquid administration on respiration.

### 4.2. Conditioned Taste Aversion Test

The experimental schedule is shown in Figure 1A. After intranasal administration of rotenone for 7 days, recovery time was set for three days to avoid the influence of administration. After the recovery, water-deprived mice for 16 h were trained for two days to perform brief-access licking of distilled water through a single drinking bottle for no more than 2 s. On the next day, the mice were presented with 0.03 mM quinine hydrochloride (QHCl; Nacalai Tesque, Kyoto, Japan) solution for 10 min, and the amount consumed from a single drinking bottle was measured to test innate aversiveness to 0.03 mM QHCl. The dose of QHCl was set near the aversion threshold in naive animals [59]. On the next two days, the paring of conditioned stimulus (CS) with the unconditioned stimulus (US) was repeated. The mice were presented with 0.03 mM QHCl solution as CS and given an intraperitoneal injection of 0.6 M lithium chloride (LiCl; Wako Pure Chemical Industries) (0.6 mg/kg body weight) as US, immediately after the vigorous drinking of the CS solution. The CS-US interval was typically less than 30 s. The mice returned to their home cages and given free access to food and water. From the next day, the amount of 0.03 mM QHCl solution consumed for 10 min was measured for 6 days, in order to examine the change of the aversiveness to 0.03 mM QHCl after conditioned taste aversion. The amount of intake of QHCl solution was normalized by body weight (ml/10 g body weight).

### 4.3. Slice Preparation

Vehicle (DMSO)-treated and rotenone-treated mice were used in the experiments. Following intranasal administration of DMSO or rotenone for 7 days, recovery time was set for 5-7 days to avoid the influence of administration. After the recovery, coronal slices, including the insular cortex, which is involved in the conditioned taste aversion memory [25,60,61] were prepared as described previously [26]. Briefly, the mice were deeply anesthetized with isoflurane and decapitated. The brain was removed rapidly from the skull and placed in cold modified artificial cerebrospinal fluid (aCSF) (in mM): 210 sucrose, 2.5 KCl, 2.5 MgSO_4_, 1.25 NaH_2_PO_4_, 26 NaHCO_3_, 0.5 CaCl_2_ and 50 D-glucose. Coronal slices, including the gustatory insular cortex were cut at 300 μm thickness using a microslicer (Linearslicer Pro 7, Dosaka EM, Kyoto, Japan). Slices were incubated at 32 °C for 30 min in a submersion type chamber, which contained 50% modified aCSF and 50% normal aCSF (pH 7.3). Normal aCSF contained (in mM) 126 NaCl, 3 KCl, 1 MgSO_4_, 1.25 NaH_2_PO_4_, 26 NaHCO_3_, 2 CaCl_2_ and 10 D-glucose. Thereafter, the brain slices were transferred to a storage chamber containing normal aCSF and maintained at room temperature until used for recordings. Normal aCSF was continuously aerated with a mixture of 95% O_2_–5% CO_2_.

### 4.4. Whole-Cell Patch-Clamp Recordings

Whole-cell patch-clamp recordings were performed in a recording chamber on the stage of a microscopy with infrared differential interference contrast optics for visualization of neurons (BX-51WI; Olympus, Tokyo, Japan). Coronal brain slices including the gustatory insular cortex were transferred to the recording chamber. The slices were perfused with normal aCSF with a flow rate of 2 mL/min. Whole-cell recordings were made from visually identified layer V pyramidal neurons of the gustatory insular cortex using MultiClamp 700B Amplifier (Molecular Devices, Foster City, CA, USA). All electrophysiological recordings were carried out at temperature of 30–32 °C.

When postsynaptic currents (PSCs) were recorded, recording pipettes (3–5 MΩ) were filled with solution containing the following (in mM): 132.5 K-gluconate, 8.5 KCl, 14 Na-gluconate, 2 ATP-Mg, 0.3 GTP-Na_3_, 10 4-(2-hydroxyethyl)-1-piperazineethanesulfonic acid (HEPES) and 0.2 ethylene glycol tetraacetic acid (EGTA); adjusted to pH 7.3 with KOH [26]. By using this internal solution, the equilibrium potential for Cl^–^ was set to –70 mV. PSCs were recorded from layer V pyramidal neurons at a holding potential of –70 mV. Under this recording condition, contamination of inhibitory PSCs was eliminated and, thus, evoked PSCs were solely mediated by excitatory synaptic inputs onto layer V pyramidal neurons in the mouse insular cortex [26]. Electrical stimuli were delivered by a monopolar tungsten stimulating electrode placed within layer V in the gustatory insular cortex (~200 µm along the apical dendrite of the neuron). PSCs were evoked by extracellular repetitive stimuli (duration is 100 μs and stimulation intensity was adjusted to induce PSCs with amplitudes of 50–100 pA at 0.033 Hz). Spontaneous excitatory PSCs (sEPSCs) were recorded in the presence of 10 μM bicuculline. When inhibitory PSCs (IPSCs) were recorded, recording pipettes (3–5 MΩ) were filled with solution containing the following (in mM): 130 Cs-gluconate, 10 CsCl, 2 MgCl_2_, 2 ATP-Na_2_, 0.4 GTP-Na_3_, 10 HEPES and 0.2 EGTA; adjusted to pH 7.3 with CsOH. The Cl^–^ equilibrium potential (E_Cl_) was set to be −57 mV [62]. IPSCs were recorded at a holding potential of 0 mV. The IPSC recordings were started 10 min after establishing the whole-cell configuration and clamping at 0 mV. Spontaneous IPSCs (sIPSCs) were recorded in the presence of 10 μM DNQX (6,7-dinitroquinoxaline-2,3-dione) and 50 μM AP-5 (DL-2-amino-5-phosphonopentanoic acid). Spike firings were evoked by current injections, which were started at 0 pA and increased in increments of 30 pA until 240 pA. The sealing resistance was usually larger than 10 GΩ. The membrane potential values were corrected for the estimated liquid junction potential (10 mV). Records of currents and voltages were low-pass filtered at 2 kHz (4-pole Bessel filter), digitized at a sampling rate of 2–10 kHz (1440A, Molecular Devices, Sunnyvale, CA, USA), and stored on a computer hard disk.

### 4.5. Induction Protocol for Synaptic Plasticity

To induce LTP in pyramidal neurons of the mouse insular cortex, 80 presynaptic stimuli were applied at 2 Hz, during which (40 s) recorded neurons were held at +30 mV [26,63]. The duration and intensity of presynaptic stimuli were the same as the control stimuli. To induce LTD, 300 presynaptic pulses were applied at 1 Hz, during which (300 s) recorded neurons were held at −45 mV [27]. Synaptic plasticity was induced with the LTP- and LTD-inducing protocol within 12 min after establishing the whole-cell configuration to avoid washout of intracellular contents that are critical for the establishment of synaptic plasticity [26,63]. To examine how muscimol affects induction of LTD in pyramidal neurons of rotenone-treated mice, muscimol was applied either during the LTD-inducing stimuli (300 s) or without them. No antagonists of inhibitory synaptic transmission were used during LTP and LTD recordings. Access resistance was 15–20 MΩ and monitored throughout the experiment. Data were discarded if access resistance changed more than 15% during experiments.

### 4.6. Drug Application for Whole-Cell Patch-Clamp Recordings

The following drugs were used for electrophysiological recordings. The NMDA receptor antagonist (AP-5), the GABA_A_ receptor agonist (muscimol), the GABA_A_ receptor antagonist (bicuculline), and the non-NMDA receptor antagonist (DNQX) were bath-applied at concentrations of 50 μM, 50 μM, 10 μM, and 10 μM, respectively. AP-5 was purchased from Tocris Bioscience (Bristol, UK). All other chemicals were purchased from Sigma–Aldrich (St Louis, MO, USA).

### 4.7. Data Analysis

Analyses of electrophysiological data were performed, similar to our previous study [26]. Synaptic plasticity was evaluated as the change in PSC amplitude when comparing an average PSC amplitude recorded for 5 min (25–30 min after LTP/LTD-inducing stimuli) to a baseline PSC amplitude recorded in the last 5 min of control responses. The effect of muscimol application alone on PSC amplitude was evaluated as the change in PSC amplitude when comparing an average PSC amplitude recorded for 5 min (25–30 min after muscimol application) to a baseline PSC amplitude recorded in the last 5 min of control responses. The frequency and amplitude of sEPSCs and sIPSCs were analyzed using MiniAnalysis program 6.0.3 (Synaptosoft, Decatur, GA, USA).

All data were expressed as the mean ± S.E. Statistical significance was assessed using unpaired or paired Student’s *t*-test and two-way repeated-measures ANOVA followed by Fisher’s protected least significant difference (LSD) post-hoc test. Where described in text, Student’s *t*-tests were used to compare responses from several neurons. Student’s *t*-test was used when the data were normally distributed. Statistical results were displayed using a precise *p* value, except when *p* was less than 0.001 (*p* < 0.001). A *p*-value less than 0.05 was considered statistically significant.

## Figures and Tables

**Figure 1 ijms-22-00259-f001:**
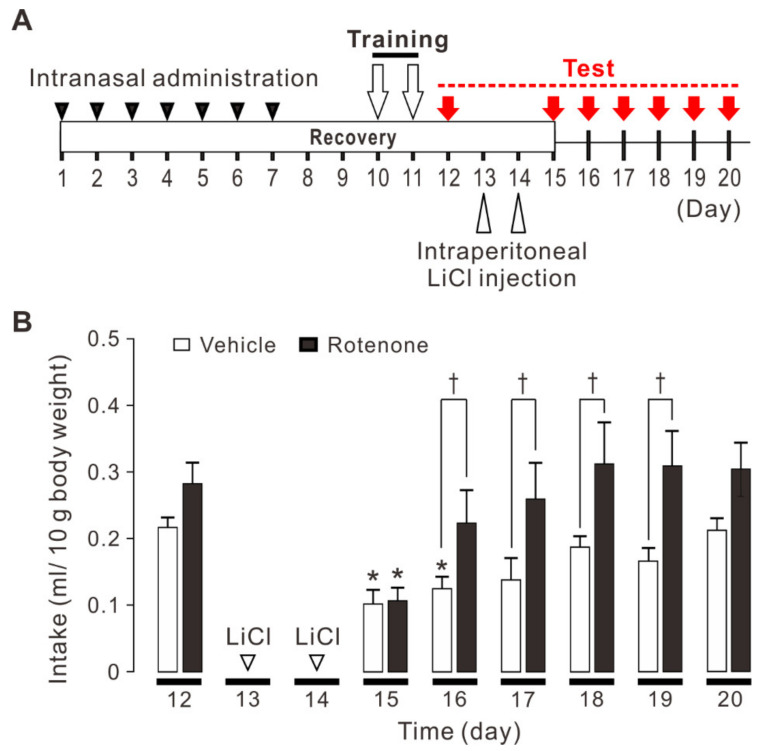
Conditioned taste aversion to quinine hydrochloride (QHCl) in vehicle (DMSO)-treated and rotenone-treated mice. (**A**) The experimental schedule in this study. After intranasal administration of rotenone for 7 days and recovery time for 3 days, water-deprived mice for 16 h were trained for two days to perform brief-access licking of distilled water through a single drinking bottle. On the next day, the mice were presented with 0.03 mM QHCl solution for 10 min and the amount consumed from a single drinking bottle was measured. Over the next two days, the paring of CS with US was repeated. The mice were presented with 0.03 mM QHCl solution as CS and given an intraperitoneal injection of 0.6 M LiCl (0.6 mg/kg body weight) as US. From the next day, the amount of 0.03 mM QHCl solution consumed for 10 min was measured for 6 days. (**B**) Both vehicle-treated mice (*n* = 6) and rotenone-treated mice (*n* = 6) showed conditioned taste aversion on the 15th day (1 day after paring of CS with US). On the 16th day (2 days after pairing of CS with US), conditioned taste aversion was extinguished rapidly in rotenone-treated mice, but not in vehicle-treated mice, returning to the control level. On the 16th, 17th, 18th, and 19th day (2, 3, 4, and 5 days after pairing of CS with US), the amount of intake was significantly greater in rotenone-treated mice than in vehicle-treated mice. Two-way ANOVA LSD: * *p* < 0.05 vs respective controls (intake on the 12th day); Two-way ANOVA LSD: † *p* < 0.05.

**Figure 2 ijms-22-00259-f002:**
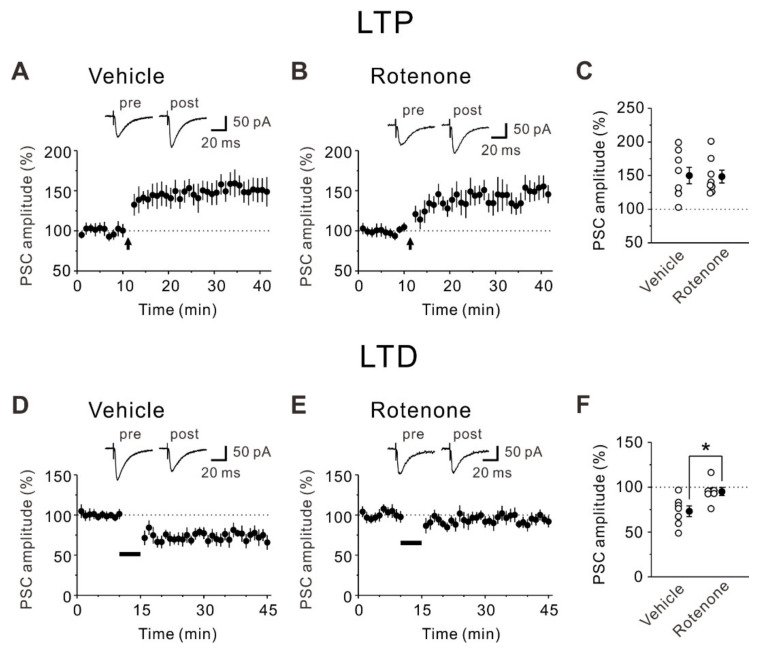
Impairment of LTD caused by intranasal administration of rotenone in layer V pyramidal neurons of the mouse insular cortex. (**A**) Synaptic potentiation was induced by LTP-inducing stimuli (presynaptic 80 pulses at 2 Hz with postsynaptic depolarization at +30 mV) in insular pyramidal neurons of vehicle-treated mice (*n* = 8). (**B**) Synaptic potentiation was induced by LTP-inducing stimuli in insular pyramidal neurons of rotenone-treated mice (*n* = 8). (**A**,**B**) The insets show averages of ten consecutive current traces before (pre) and 25–30 min after LTP-inducing stimuli (post). The dashed line indicates the mean basal synaptic responses. LTP-inducing stimuli are indicated by an arrow. (**C**) Summary scatter plots of normalized PSC amplitude obtained in (**A**,**B**). (**D**) Synaptic depression was induced by LTD-inducing stimuli (presynaptic 300 pulses at 1 Hz with postsynaptic depolarization at −45 mV) in insular pyramidal neurons of vehicle-treated mice (*n* = 7). (**E**) Synaptic depression was not induced by LTD-inducing stimuli in insular pyramidal neurons of rotenone-treated mice (*n* = 7). (**D**,**E**) The insets show averages of ten consecutive current traces before (pre) and 25–30 min after LTD-inducing stimuli (post). The dashed line indicates the mean basal synaptic responses. LTD-inducing stimuli are indicated by a solid bar. (**F**) Summary scatter plots of normalized PSC amplitude obtained in (**D**,**E**). Unpaired *t*-test, *: *p* < 0.02.

**Figure 3 ijms-22-00259-f003:**
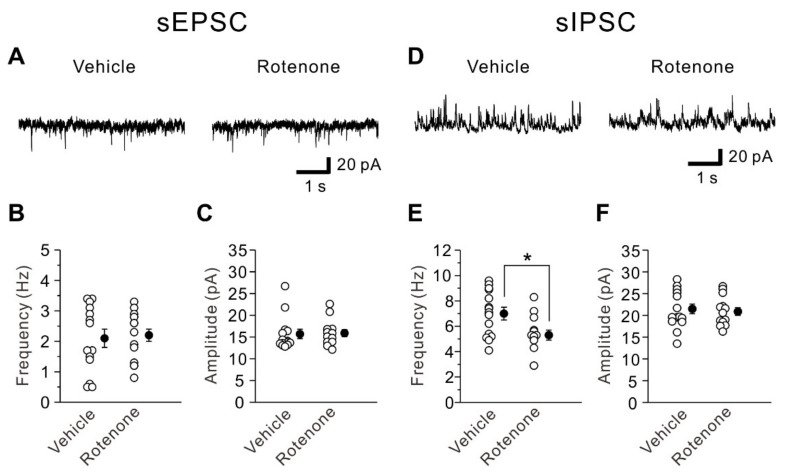
Impairment of inhibitory synaptic transmission caused by intranasal administration of rotenone. (**A**) Representative traces of sEPSCs obtained from layer V pyramidal neurons of vehicle-treated mice (left) and rotenone-treated mice (right). (**B**,**C**) Summary scatter plots of sEPSC frequency (**B**) and amplitude (**C**) obtained from vehicle-treated mice (*n* = 14) and rotenone-treated mice (*n* = 14). (**D**) Representative traces of sIPSCs obtained from layer V pyramidal neurons of vehicle-treated mice (left) and rotenone-treated mice (right). (**E**,**F**) Summary scatter plots of sIPSC frequency (**E**) and amplitude (**F**) obtained from vehicle-treated mice (*n* = 15) and rotenone-treated mice (*n* = 13). Unpaired *t*-test, *: *p* < 0.02.

**Figure 4 ijms-22-00259-f004:**
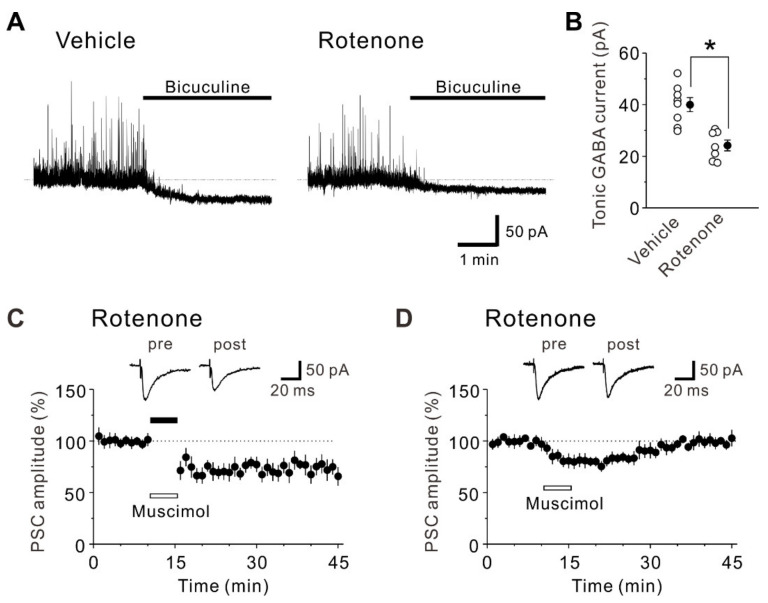
Reduction of tonic GABA currents caused by intranasal administration of rotenone and restoration of LTD by muscimol in pyramidal neurons of rotenone-treated mice. (**A**) Representative current traces recorded before and during application of 10 μM bicuculline in pyramidal neurons obtained from vehicle-treated mice (left) and rotenone-treated mice (right). (**B**) Summary scatter plots of tonic GABA currents obtained from pyramidal neurons of vehicle-treated mice (*n* = 8) and rotenone-treated mice (*n* = 7). Unpaired *t*-test, *: *p* < 0.001. (**C**) LTD was induced by LTD-inducing stimuli in pyramidal neurons of rotenone-treated mice when 50 μM muscimol was present (*n* = 8). LTD-inducing stimuli are indicated by a solid bar. The insets show averages of ten consecutive current traces before (pre) and 25–30 min after the LTD-inducing stimuli (post). (**D**) LTD was not induced by muscimol application alone in pyramidal neurons of rotenone-treated mice (*n* = 7). The insets show averages of ten consecutive current traces before (pre) and 25–30 min after muscimol application (post).

**Figure 5 ijms-22-00259-f005:**
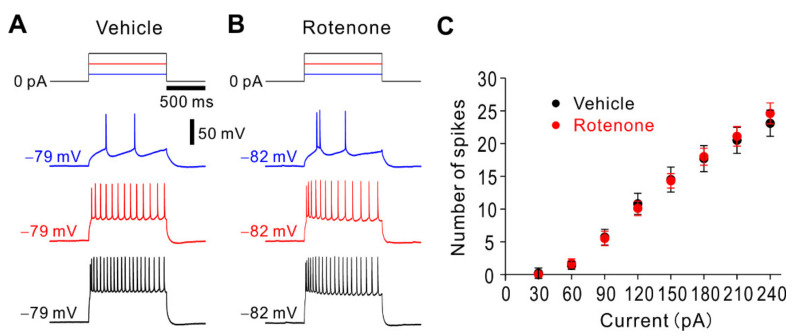
Effects of intranasal administration of rotenone on spike firing properties of layer V pyramidal neurons. (**A**,**B**) Representative spike firings obtained from vehicle-treated mice (**A**) and rotenone-treated mice (**B**). Spike firings were evoked by current injection at 60 pA (blue traces), 150 pA (red traces) and 240 pA (red traces), respectively. (**C**) Relationship between the injected current intensity and number of spikes obtained from vehicle-treated mice (*n* = 16) and rotenone-treated mice (*n* = 14).

## Data Availability

The data presented in this study are available in request from the corresponding author.

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
