# Peer review of "Intranasal Administration of Rotenone Reduces GABAergic Inhibition in the Mouse Insular Cortex Leading to Impairment of LTD and Conditioned Taste Aversion Memory"

_ijms, 2020, doi:10.3390/ijms22010259_

Round 1
Reviewer 1 Report
The authors, focusing their attention on the pesticide rotenone ,whose toxic effects are known to cause Parkinson's disease and cognitive impairment, have successfully demonstrated that the pesticide induces LTD. In fact it is capable of producing a decrease in the inhibitory currents of the GABA-ergic system, the only mammalian inhibitory system, in pyramidal neurons (of layer V) of the insular cortex. This in turn produces an impairment of all other synaptic transmission such as serotonergic, cholinergic and dopaminergic systems, thus demonstrating that this latter system is not the principal target of the cytotoxic activity of rotenone, as previously reported in literature. In fact, it is possible to revert the effect of rotenone by means of the GABA-A agonist, muscimol. Furthermore, in a very interesting way, they investigated the toxic effects of rotenone on cognitive abilities through the test CTA, a well-established learning and memory paradigm and considered a special form of conditioning, closely related to the activity of the amygdala, where it is known that GABA-ergic transmission interneurons are present. The chosen administration is the inhalation one which is usually the way through which rotenone, taken accidentally, exerts its neurotoxic effects; in fact, administered intranasally in the mice, because of its high lipophilicity, it is able to cross the BBB, directly reach the brain through an active axonal transport of olphactory neurons and providing its toxic effects. From the point of view of pharmacological investigation, this manuscript well describes behaviours and neural processes associated with CTA and provides an important contribution to elucidate the role GABA-ergic system in cognitive decline.
Author Response
Thank you very much for the positive comment.
We really appreciate it.
Reviewer 2 Report
The present study by Toyoda & Sato shows that physiologically intranasal administration of rotenone impairs conditioned taste aversion. This is associated with a reduction in phasic/tonic GABAergic transmission and an impairment in the induction of long-term synaptic depression. This is a nice study that examines the effects of a physiologically-relevant route of rotenone administration and demonstrates synaptic changes that contrast those found in other brain regions. I just have one major comment and a number of minor comments which I hope will improve the integrity of this study:
- One of the key findings is that augmentation of GABA transmission with the GABAA receptor agonist, muscimol, restores the ability to induce long-term depression (LTD). This enhancement of GABA transmission with muscimol occurs during the 5 min induction phase of LTD (Fig. 4C). However, there is evidence suggesting that transient muscimol application can induce LTD alone (eg. Zhang et al., 2016). Thus, there is the possibility that the rescue of LTD observed is the result of muscimol application rather than the induction protocol. Perhaps a control for this would be to examine the effect of 5 min muscimol application alone (without 300 pulses @ 1 Hz). A transient effect by muscimol would bolster the current claims, while a long-lasting effect would simply change the narrative.
- While one of the key findings is that GABAergic transmission is reduced, the impairment in LTD is of glutamatergic transmission. Yet there is very little mention in the manuscript of this discrepancy. Could you please elaborate more on how a GABA inhibition facilitates LTD of glutamatergic transmission? There is much existing literature regarding heterosynaptic plasticity between GABA and glutamate. I believe it would greatly improve the manuscript by clarifying this connection.
- Line 420-421 (Methods): “No blockers of GABAergic transmission were used during LTP and LTD recordings.”
Does this mean both inhibitory an excitatory postsynaptic currents (IPSCs and EPSCs) were recorded during the LTP/LTD experiments? If so, the use of “EPSC amplitude” in the figures and text is slightly deceiving. Perhaps rewording to a less specific term (such as PSC) may be more accurate? The answer to this query may also help clear up the point made directly above this.
- In contrast to existing literature from other brain regions, the present study finds that rotenone produces no change in EPSCs and LTP. This is an interesting observation, but also brings into question why the insular cortex is different. Perhaps it would improve the manuscript by postulating why the insular cortex might be unique to such changes? Or is there any evidence suggesting that intranasal administration might be more specific to GABAergic over glutamatergic neurons?
- Line 283-285: “Because dopamine neurons in the glomerular layer of the olfactory bulb are inhibitory neurons for the mitral cells [39], these data strongly suggest that the reduced inhibitory inputs by rotenone are caused by the degeneration of dopamine neurons”
- Line 287-291: “Since it has been shown that the intranasal administration of rotenone caused degeneration of the substantia nigra dopaminergic neurons, the rotenone-induced reduction of inhibitory synaptic inputs onto insular layer V pyramidal neurons is likely to be induced by the degeneration of the substantia nigra dopaminergic neurons.”
Both statements above speculating that dopaminergic degeneration likely mediates the GABAergic deficits is interesting, but comes off a bit too strong. One way to confirm this would be to examine if augmentation of dopaminergic transmission rescues LTD. However, rather than doing more experiments, it would be simpler to reword these sentences and temper the conclusion slightly (eg. perhaps reword “strongly suggest”, and is “likely to be induced”).
- Fig. 3B: The sample trace shown in this figure is not representative of the average IPSC amplitude shown in Fig. 3D. Furthermore, while I realize the scale bars are different between the traces in Fig. 3A and 3B, one still looks quite different from the other. Can the authors possibly provide more consistent, representative traces?
- Figs. 5A/5B: The sample neuron shown for Vehicle is held at -69 mV, while the sample neuron for Rotenone is held at -72 mV. Were the current injections performed with the neurons held at the same holding potential, or simply from resting membrane potential? Either is OK, but perhaps this should be more explicitly defined in the methods and/or figure legend.
REFERENCES:
Zhang X, Yao N, Chergui K. The GABAA receptor agonist muscimol induces an age- and region-dependent form of long-term depression in the mouse striatum. Learn Mem 2016; 23: 479–485.
Author Response
Responses to Reviewer #2
We thank the reviewer #2 for the suggestions to improve the manuscript. Following the suggestions, we have revised the manuscript.
1. One of the key findings is that augmentation of GABA transmission with the GABAA receptor agonist, muscimol, restores the ability to induce long-term depression (LTD). This enhancement of GABA transmission with muscimol occurs during the 5 min induction phase of LTD (Fig. 4C). However, there is evidence suggesting that transient muscimol application can induce LTD alone (eg. Zhang et al., 2016). Thus, there is the possibility that the rescue of LTD observed is the result of muscimol application rather than the induction protocol. Perhaps a control for this would be to examine the effect of 5 min muscimol application alone (without 300 pulses at 1 Hz). A transient effect by muscimol would bolster the current claims, while a long-lasting effect would simply change the narrative.
Reply: We appreciate this comment. According to this suggestion, we have examined whether or not muscimol application alone (without 300 pulses at 1 Hz) can induce LTD in layer V pyramidal neurons of the insular cortex in rotenone-treated mice. We found that muscimol application alone did not induce LTD (new Fig. 4D), indicating that the effect of muscimol is transient and the restoration of LTD is required for not only muscimol but also LTD-inducing stimuli. We have added the description regarding this result in the revised manuscript (Lines 196-205).
2. While one of the key findings is that GABAergic transmission is reduced, the impairment in LTD is of glutamatergic transmission. Yet there is very little mention in the manuscript of this discrepancy. Could you please elaborate more on how a GABA inhibition facilitates LTD of glutamatergic transmission? There is much existing literature regarding heterosynaptic plasticity between GABA and glutamate. I believe it would greatly improve the manuscript by clarifying this connection.
Reply: Thank you for the suggestion. According to this suggestion, we have added the discussion in the revised manuscript as follows: “Our data suggests that the decreased GABAergic synaptic transmission by rotenone is likely to cause impairment of LTD because activation of GABAA receptors by muscimol during LTD-inducing stimuli restored LTD in pyramidal neurons of the rotenone-treated mice. Importantly, while our finding shows that the GABAergic synaptic transmission is reduced by rotenone, the impairment of LTD is brought about by an alteration of glutamatergic synaptic transmission. In other words, GABA inhibition is involved in modulation of LTD of glutamatergic synapses. The GABAergic inhibition has been shown to play a critical role in induction of glutamatergic LTD in several brain regions [29, 52-54]. For instance, it has been demonstrated that glutamatergic LTD induced by low frequency stimulation was abolished by GABAA receptor antagonists in the mouse amygdala [52] and the rat hippocampus [54]. These studies postulate that local feed-forward and/or feed-back circuitry mediated by GABAergic interneurons might be involved in induction of LTD. In the neocortex, fast-spiking basket interneurons primary mediate feed-forward and feed-back inhibition within layer V [55]. Through this circuit, LTD of glutamatergic synapses might be induced in layer V pyramidal neurons of the mouse insular cortex. Thus, the reduction of dendritic GABAergic inhibition in rotenone-treated mice appears to shift the threshold for induction of LTD, resulting in an impairment of LTD. In addition, it has been shown that application of muscimol induced LTD in the rat hippocampus [53, 54], the nucleus accumbense of adolescent mice [29] and the dorsolateral striatum of adult mice [29]. With regard to a possible mechanism for this type of synaptic modification, both presynaptic and postsynaptic mechanisms are likely to be involved. In particular, it has been shown that following application of muscimol, the release of endocannabinoids which mediate retrograde signals occurred presynaptically to reduce release of glutamate in the striatum complex in an age- and region- dependent manner [29]. Thus, muscimol-mediated LTD is believed to be caused by retrograde messengers. However, we have demonstrated that muscimol application alone did not cause LTD in insular pyramidal neurons of rotenone-treated mice. Therefore, the latter possibility seems to be unlikely.” (Lines 339-363)
3. Line 420-421 (Methods): “No blockers of GABAergic transmission were used during LTP and LTD recordings.”
Does this mean both inhibitory an excitatory postsynaptic currents (IPSCs and EPSCs) were recorded during the LTP/LTD experiments? If so, the use of “EPSC amplitude” in the figures and text is slightly deceiving. Perhaps rewording to a less specific term (such as PSC) may be more accurate? The answer to this query may also help clear up the point made directly above this.
Reply: We appreciate this comment. As suggested by the reviewer, both inhibitory and excitatory postsynaptic currents (IPSCs and EPSCs) were recorded during the LTP/LTD experiments. Thus, we have changed from “EPSC amplitude” to “PSC amplitude” in the text and figures.
4. In contrast to existing literature from other brain regions, the present study finds that rotenone produces no change in EPSCs and LTP. This is an interesting observation, but also brings into question why the insular cortex is different. Perhaps it would improve the manuscript by postulating why the insular cortex might be unique to such changes? Or is there any evidence suggesting that intranasal administration might be more specific to GABAergic over glutamatergic neurons?
Reply: We appreciate this comment. Currently, it is unclear why the insular cortex is different from other brain regions. We had already discussed the possible explanation of our results in the original manuscript (Lines 283-292). Alternatively, we have added the discussion about the vulnerability to rotenone in GABAergic and glutamatergic neurons as follows: “Based on our results, the intranasal administration of rotenone produced almost no effect on glutamatergic synaptic transmission while reducing GABAergic synaptic transmission in the mouse insular cortex. It is currently unclear whether the effect of the intranasal administration of rotenone is more specific to GABAergic neurons than glutamatergic neurons. The vulnerability to rotenone has previously been shown to be almost similar between cultured GABAergic and glutamatergic neurons [39, 44]. However, GABAergic neurons appeared less elongated after 100 nM rotenone treatment and thus it is indicated that GABAergic neurons are more susceptible to rotenone-induced oxidative stress [44]. In addition, rotenone has been shown to cause nitrosative stress [45], and it has been demonstrated that GABAergic neurons which express neuronal nitric oxidase synthase (nNOS) are more susceptible to toxic exposure due to their intrinsic high basal levels of nitrosative stress in comparison with glutamatergic neurons [46]. Thus, it is possible that rotenone may cause functional impairment in nNOS-expessing GABAergic neurons more easily than in glutamatergic neurons. Further studies of the precise mechanisms as to how the intranasal administration of rotenone affects GABAergic and glutamatergic neurons in the insular cortex would be necessary.” (Lines 308-321).
5. Line 283-285: “Because dopamine neurons in the glomerular layer of the olfactory bulb are inhibitory neurons for the mitral cells [39], these data strongly suggest that the reduced inhibitory inputs by rotenone are caused by the degeneration of dopamine neurons”
Line 287-291: “Since it has been shown that the intranasal administration of rotenone caused degeneration of the substantia nigra dopaminergic neurons, the rotenone-induced reduction of inhibitory synaptic inputs onto insular layer V pyramidal neurons is likely to be induced by the degeneration of the substantia nigra dopaminergic neurons.”
Both statements above speculating that dopaminergic degeneration likely mediates the GABAergic deficits is interesting, but comes off a bit too strong. One way to confirm this would be to examine if augmentation of dopaminergic transmission rescues LTD. However, rather than doing more experiments, it would be simpler to reword these sentences and temper the conclusion slightly (eg. perhaps reword “strongly suggest”, and is “likely to be induced”).
Reply: We appreciate this comment. As suggested by the reviewer, we have reworded these sentences and tempered the conclusion slightly as follows:
Because dopamine neurons in the glomerular layer of the olfactory bulb are inhibitory neurons for the mitral cells [40], these data may suggest that the reduced inhibitory inputs by rotenone are caused by the degeneration of dopamine neurons [13].(Lines 296-298)
Since it has been shown that the intranasal administration of rotenone caused degeneration of the substantia nigra dopaminergic neurons, the rotenone-induced reduction of inhibitory synaptic inputs onto insular layer V pyramidal neurons is possibly induced by the degeneration of the substantia nigra dopaminergic neurons. (Lines 300-304)
6. Fig. 3B: The sample trace shown in this figure is not representative of the average IPSC amplitude shown in Fig. 3D. Furthermore, while I realize the scale bars are different between the traces in Fig. 3A and 3B, one still looks quite different from the other. Can the authors possibly provide more consistent, representative traces?
Reply: Thank you for the suggestion. As suggested by the reviewer, we have replaced the sample sIPSC traces in Fig. 4D.
7. Figs. 5A/5B: The sample neuron shown for Vehicle is held at -69 mV, while the sample neuron for Rotenone is held at -72 mV. Were the current injections performed with the neurons held at the same holding potential, or simply from resting membrane potential? Either is OK, but perhaps this should be more explicitly defined in the methods and/or figure legend.
Reply: Thank you for the suggestion. The current injections were applied from the resting membrane potential. In the methods, we have added this information as follows: “Spike firings were evoked by current injections which were started at 0 pA and increased in increments of 30 pA until 240 pA” (Lines 455-456). In addition, we modified Fig. 5A and 5B, in which “0 pA” was added in the beginning of the current injection.

Round 2
Reviewer 2 Report
I thank the authors for their thorough response and revisions. They have sufficiently addressed all concerns that I previously raised, and I have no further suggestions. Great job with this study.